# Canine Melanomas as Models for Human Melanomas: Clinical, Histological, and Genetic Comparison

**DOI:** 10.3390/genes10070501

**Published:** 2019-06-30

**Authors:** Anaïs Prouteau, Catherine André

**Affiliations:** Univ Rennes, CNRS, IGDR - UMR 6290, F-35000 Rennes, France

**Keywords:** melanoma, dogs, spontaneous model, clinics, histology, genetics, comparative oncology

## Abstract

Despite recent genetic advances and numerous ongoing therapeutic trials, malignant melanoma remains fatal, and prognostic factors as well as more efficient treatments are needed. The development of such research strongly depends on the availability of appropriate models recapitulating all the features of human melanoma. The concept of comparative oncology, with the use of spontaneous canine models has recently acquired a unique value as a translational model. Canine malignant melanomas are naturally occurring cancers presenting striking homologies with human melanomas. As for many other cancers, dogs present surprising breed predispositions and higher frequency of certain subtypes per breed. Oral melanomas, which are much more frequent and highly severe in dogs and cutaneous melanomas with severe digital forms or uveal subtypes are subtypes presenting relevant homologies with their human counterparts, thus constituting close models for these human melanoma subtypes. This review addresses how canine and human melanoma subtypes compare based on their epidemiological, clinical, histological, and genetic characteristics, and how comparative oncology approaches can provide insights into rare and poorly characterized melanoma subtypes in humans that are frequent and breed-specific in dogs. We propose canine malignant melanomas as models for rare non-UV-induced human melanomas, especially mucosal melanomas. Naturally affected dogs offer the opportunity to decipher the genetics at both germline and somatic levels and to explore therapeutic options, with the dog entering preclinical trials as human patients, benefiting both dogs and humans.

## 1. Introduction

Human malignant melanoma is a highly heterogeneous cancer that affects numerous anatomical sites with different clinical behaviours. It originates from melanocytes, which are dendritic cells located in epithelia such as skin or mucosa, but also in the eye or internal organs, thus defining major subtypes with specific anatomic locations and pathogenesis. The major subtypes include cutaneous (UV-induced and non-UV-induced types), acral, mucosal, and ocular melanomas. Epidemiology, UV-radiation status, histopathological features, genetics, prognosis, and outcomes vary considerably between these subtypes. 

Cutaneous melanomas account for approximately 75% of skin cancer deaths [1,2], and their incidence is rising among populations of European descent, with 80,000 cases per year in the USA [3]. The current World Health Organization (WHO) classification for cutaneous melanoma is based on morphologic aspects of the early growth phase and the body site of the primary melanoma, to distinguish four main types: superficial spreading melanoma (SSM), lentigo maligna melanoma (LMM), nodular melanoma (NM), and acral lentiginous melanoma (ALM) [1,2,3,4,5]. Chronic and intermittent ultraviolet exposures are a major cause of melanoma development in sun-exposed skin. UV-induced melanoma with cumulative sun-induced damage (CSD) and non-CSD melanoma are the most common types in peoples of European descent and occur at around 60 and 40 years of age, respectively [1]. Another type of melanoma affecting non-sun-exposed or specific sites is acral melanoma, with locations on the palms of the hands, soles of the feet, and the nail apparatus, which occurs at similar incidences across all world populations [6]. Additional rare categories of melanomas present distinct clinical, epidemiological, and histopathological features, such as desmoplastic melanoma, spitzoid melanoma, blue nevus-like melanoma, melanoma in giant congenital nevus, and other types arising from dermal melanocytes (nevus of Ota, nevus of Ito) [1]. Even if the major part of cutaneous melanoma is diagnosed at an early stage, approximately 10% of melanoma cases are unresectable or already metastatic at diagnosis [7]. For those latter patients, the prognosis is poor, with a 5 year survival rate of around 12% [8], whereas it is around 90% for cutaneous melanomas of all stages [9,10]. Acral subtypes carry the worst prognosis among cutaneous melanomas, with a 5 year survival rate of around 80% [11]. The impact of other subtypes on prognosis is not known [5]. 

Mucosal melanomas originate from melanocytes within mucosal epithelia and constitute a rare clinical entity representing 0.8–3.7% of all melanomas [12], with an incidence of 800 cases per year in the USA [3]. Although rare in the general population, these account for 9–22% of melanoma cases in Asian and black populations where cutaneous melanoma is less common [12] and occur around the age of 60–70 [3]. They exhibit a pathogenic mechanism different from cutaneous subtypes and a much worse prognosis. They can originate from the anogenital regions, oropharynx and paranasal sinuses, as well as the conjunctiva (head and neck localisations), and can also arise from parotid glands, oesophagus, and the middle ear [13]. Local and/or distant metastasis to lungs and liver occur in 50% of cases with head and neck melanoma [3]. Mucosal melanomas carry a poor prognosis, with a 5 year survival rate between 25% and 33%, according to disease stage and location [3,14].

Uveal melanoma is the most common intraocular human cancer and accounts for approximately 5% of all melanoma in the United States [1], occurring at around 60 years of age [15]. The incidence of uveal melanoma is stable with 2500 new cases per year in the USA [3]. The most frequent sites are the choroid and the ciliary body. Although uveal melanoma is rare, the disease is still lethal. Almost half of patients with effective primary therapy will develop metastases [16], and these patients have a poor prognosis, with a median survival time ranging from 4 to 15 months [17]. 

## 2. Comparative Oncology, Model Specificities

Melanomas are generally aggressive, with a high metastatic propensity toward several organs including lungs, bones, liver, and the central nervous system. Even though the development of targeted therapy and immunotherapy has improved outcomes, there is still a lack of knowledge to enable the development of efficient treatments with sustainable response and to provide clues to the mechanisms of resistance. Despite numerous efforts to better understand the genetics and physiopathology of melanomas, previous studies mainly focused on cutaneous melanoma and often considered different melanoma types as a single disease entity. Fortunately, recent published data in the discipline has highlighted and precisely described molecular specificities based on the different melanoma subtypes.

The discovery of genetic alterations has led to the development of targeted therapies, such as tyrosine kinase inhibitors or BRAF inhibitors, new therapeutic options, and stratification of patients. However, melanoma research could certainly benefit from additional models to those currently used. Indeed, the popular mouse model used over the last decades in the study of cancer has revealed some limits: patient-derived xenograft mouse models lack a fully functional immune system, and genetically engineered mouse models are expensive and do not carry the same mutagenic load and genetic background as human cancers [18]. It is also now recognized that mice were not as efficient as expected to objectify drug response in Phase III clinical trials. Researchers then became more and more interested in naturally occurring cancers in pet animals, because they are immuno-competent individuals sharing our environment, and the natural history of cancer in animals occurs in a generally shorter period than in humans. Thus, many cancers in pet dogs show similar features to their human counterparts, including clinical and histological appearance, biological behaviour, tumor genetics, molecular pathways and targets, and response to therapies [19,20,21,22]. Most recent genetic analyses of cancer in pet dogs has shown their usefulness in specifically identifying key somatic genes and relevant mutational mechanisms [22,23,24,25,26,27,28,29,30]. Their advantages require no further demonstration and, considering the ethical constraints in the use of experimental models in respect of the 3R rules (Replace, Refine, Reduce), the use of naturally affected pets in their environment and followed-up by their veterinarians should be encouraged in comparative oncology programs.

For melanomas, obvious unmet needs remain to better manage, treat, and prevent this disease, considering their numerous specific subtypes. To this end, recent reviews have demonstrated the strength of spontaneous canine melanomas as relevant genetic and therapeutic models for specific human melanoma subtypes [18,21,31,32,33,34,35].

## 3. Canine Melanomas

Melanocytic tumors are quite common in the dog, and most of them are malignant [34,36,37,38,39,40]. Of over 2350 cases of melanocytic tumors, 70% were malignant melanomas and 30% benign tumors (called melanocytomas) [34]. Melanoma represents 7% of all malignant tumors in dogs [37], and the most frequent location is the oral cavity, mimicking human mucosal melanomas [32,33,34]. Canine melanomas also occur in the haired skin, nail bed, footpad, eye (Figure 1), gastro-intestinal tract, central nervous system, or muco-cutaneous junction [37,38,41], and it is well known that prognosis greatly depends on tumor location (Figure 2 and Figure 3); oral melanomas presenting the worse prognosis with a median survival time (MST) of 6 months, whereas MST was not reached in cutaneous forms. Digital forms seem to have intermediate prognosis, with an MST of 11.8 months [30,32,33,34,35,36,37]. Melanoma is a tumor of older dogs, with no sex predilection, but some breeds are more prone to develop a specific melanoma subtype [18,34,42,43,44]. Contrary to human cutaneous melanoma, which are the most frequent and for which UV light exposure and fair skin are known to be important etiological factors, canine melanoma etiology is not known. However, UV light exposure is not thought to be an etiological factor. Indeed, dog skin is covered with fur, and dogs with dark coat colours are more prone to develop melanoma. Most canine melanomas arise from the dermis, but melanocytes and melanocyte precursors in the dog skin are located in both layers: in the basal layer of the epidermis and in the dermis. In dogs, of course, the presence of melanocytes in the numerous hair follicles of the dermis is to be taken into consideration, but is not fully explored so far. 

A major interest with dog melanomas is that specific breeds are at higher risk to develop specific melanomas subtypes, and thus the breed is an etiological factor underlying genetic predisposition differing from breed to breed. Altogether, striking homologies have been observed between canine melanomas and human non-UV-induced melanomas; mostly oral and non-UV-induced cutaneous melanomas, like acral melanomas, also observed in people with a dark phototype [11,12].

Concerning histopathology, there are also striking homologies. Using a comparative approach, we previously reviewed 153 melanocytic tumors from oral, cutaneous, and ocular locations, and found four categories resembling human histological subtypes [34]: animal-type melanoma, melanoma simulating nevus, melanoma with epithelioid cells, and melanoma with composite cells (Figure 1) [32,33,34,39,41,42]. 

Thus, canine melanomas constitute relevant spontaneous models for human mucosal, acral, uveal, and non-UV-induced cutaneous melanomas. 

In this review, we attempt to provide the key contributions from canine melanomas in the context of comparative oncology. We present state-of-the-art epidemiology, pathology, genetics, and therapy of canine melanomas based on the different subtypes: mucosal, cutaneous, digital, and uveal melanomas.

## 4. Canine Mucosal Melanoma

### 4.1. Epidemiology 

Canine oral melanoma is the most common type of mucosal melanoma in dogs and the most common oral malignancy in this species. It occurs mostly on the gingiva, but also on the lips, tongue, tonsils, palate, and oropharynx [34,37,38,41,42]. Mean age at diagnosis is around 10–11 years of age, and several studies have identified breeds more prone to develop this oral form, such as poodles, golden and Labrador retrievers, Rottweilers, Yorkshire terrier, cocker spaniels, chow-chows, Scottish terriers, and daschunds (Figure 4A) [34,38,39,40,42,43,44]. Other mucosal sites comprise the anorectal region [45], intestines [46], and nasal cavity, but these entities are very rare in dogs [47,48,49]. 

### 4.2. Clinical Signs and Biological Behaviour

The clinical presentation of canine melanomas varies considerably; it can consist of small brown-to-black masses, but can also appear as large, flat, and/or wrinkled masses, highly to slightly pigmented or amelanic (Figure 4B). Clinical signs of oral melanoma include dysphagia, halitosis, ptyalism, bleeding, and occasionally fracture of the mandible in case of bone invasion [37,38]. This is an aggressive tumor with a local invasiveness and a high metastatic propensity to regional lymph nodes and lungs [32,33,35]. Recent studies suggest that the prevalence of pulmonary metastasis in canine oral melanoma is around 17%–51%, highlighting the aggressiveness of this disease [50,51,52]. Some cases of intranasal or anal sac melanoma have been reported, but these are rare and prognosis is poor. In a recent study, 12 anal sac melanoma cases were treated with surgery most of the time with or without adjuvant chemotherapy, and 10/12 dogs experienced local recurrence or metastasis to regional lymph nodes, penis, or lungs [45]. 

### 4.3. Histopathology 

Previous histopathological characterization of canine oral melanoma showed several major cell types, such as epithelioid cells, spindle cells, mixed types containing both patterns, and less frequent balloon or signet ring cells [36,53]. These histological types were not found to be prognostic [54]. One third of canine melanomas are amelanotic, and immunohistochemistry is sometimes needed to differentiate them from carcinomas or sarcomas [31]. Recently, oral melanoma cases with an infrequent cancer-related death rate were characterized by a limited size, an intense pigmentation, a low mitotic index, and a lack of cellular atypia, representing up to 10% of oral melanomas [33,55]. 

### 4.4. Treatment 

The first line and most effective local treatment is surgical excision [38]. Oral melanomas which invade or are in proximity to the bones are treated with partial mandibulectomy/maxillectomy, and these interventions, while highly invasive, are usually well tolerated [51,52,56,57,58]. 

Radiation therapy is generally used as a palliative treatment for unresectable tumors [59], as a neoadjuvant therapy before surgery, or as an adjuvant therapy when surgical margins are incomplete. The reported ranges of partial and complete responses to radiation therapy are 25–44% and 41–69%, respectively, yielding overall response rates of 82% to 94%, according to the studies [50,60,61,62,63]. In spite of the evident radiosentivity of oral melanoma, there is no standard treatment indicating the number of fractions and dose per fraction. 

The aggressiveness and high metastatic propensity of canine oral melanoma make systemic therapies very interesting to extend survival and slow or prevent metastases occurrence. Several studies used systemic chemotherapy with carboplatin, cisplatin, melphalan, or temozolomide [40,50,60,62,64,65,66] and showed conflicting results, but the majority of them concluded that there were no differences in terms of survival time or time to progression compared with surgery or radiation therapy. 

Thirteen years ago, Bergman et al. developed a xenogenic human tyrosinase DNA vaccine for dogs with oral melanoma, and a Phase I trial demonstrated that some dogs with advanced stage disease (presence of distant metastases) experienced long-term survival [43,67]. Since then, this DNA vaccine has been licensed by USDA (United States Department of Agriculture, Oncept TM), but further studies evaluating its efficacy reported controversial results (reviewed in Atherton et al. (2016)) [43,68,69,70]. More recent studies showed extended survival times, but the heterogeneity of inclusion criteria makes interpretation difficult [71,72,73]; in particular, a clinical response in 8/13 dogs with macroscopic tumors was reported by Verganti et al. (2017) [73].

### 4.5. Prognosis 

In canine melanomas, location is the major prognostic factor (Figure 2). Indeed, the oral subtype carries the worse prognosis, with reported median survival times (MST) between 3 months and 24 months, according to the stage (modified WHO staging system from I to IV, with tumor size, lymph node involvement, and metastases) and treatment [33,34,45,46,52,59,60,65,68,69,70]. The MST for dogs with surgically resected stage II–III oral melanoma is only 3–12 months, with an estimated one year survival rate of only 20% [70,74]. Even with a radical surgical excision, recurrence rates are still high (22–48%) [50,51,52,56,57], underlying the need for efficient systemic therapies. Regardless of the clinical stage, a threshold for Ki67 above 19.5 % predicted melanoma-related death within a year of diagnosis [75]. Among other pathological criteria, low pigmentation, a high nuclear atypia score, and a high mitotic index (>4 mitosis figures over ten fields) are linked to a poor outcome [33,37]. Most cases of oral melanoma carry a very poor prognosis, however, a small proportion of well-differentiated, slowly growing tumors has been described [52,55]. Such entities also exist in humans, for who the prognosis criteria are mainly based on histopathological features, ulceration, and Breslow parameters, but they do not constitute a given subtype.

### 4.6. Genetic Features and Comparative Aspects

Canine mucosal melanomas have been recently studied at the molecular level. Obviously different from human cutaneous melanomas, the main genetic alterations do resemble that of human mucosal melanomas. 

First analysed through candidate gene approaches, mutations were reported on *NRAS*, at the same hotspots as in human cancers (*NRAS*^Q61^), and on *PTEN,* but not on *BRAF*^V600^ and *KIT* [34,76,77]. Then, using array-comparative genomic hybridization (aCGH), recurrent deletions including *CDKN2A* and *PTEN* tumor suppressor genes were described [78]. Substantial recurrent gains of chromosomes, notably, CFA13 (*Canis familiaris*) and CFA17 (involving at least *KIT* and *MYC* oncogenes) were identified, the most recurrent aberrations being a complex copy number profile on CFA30 with alternate losses and gains and CFA 10 rearrangements encompassing the oncogenes *MDM2* and *CDK4* [78]. Most recent findings from NGS methods on canine oral melanoma of several breeds, through whole genome sequencing of five cases and targeted sequencing of 26 cases [29], exome sequencing of 65 FFPE (formalin fixed and paraffin embedded) cases [30], and exome sequencing of 69 cases [79] showed concordant results: mutations on *NRAS*, *KRAS*, occurred in 10–20% of cases, *PTEN* and *TP53* in 10 % and 8–28% of cases, respectively [29,30].

Importantly, canine oral melanoma presents numerous copy number alterations (CNA) and a relatively low single nucleotide variation (SNV) rate with general non-UV mutation signatures, as first shown in human mucosal melanomas [9]. The “chromotripsis like” profile on CFA30 was reported in canine oral melanomas concerning 60% of affected dogs and corresponds to the HSA15 alteration observed in human cases [80]. SNV detected in canine oral melanomas are also present in human mucosal melanoma [30,81,82], with exceptions for *SF3B1*, *ATRX,* and *SPRED1* mutations, that were not identified in dogs [30,83]. Conversely, recurrently altered genes such as *PTPRJ*, suggested to be inactivated in 23% of canine mucosal melanomas are not yet found in human cancers [29]. Indeed, in human mucosal melanoma cases, mutation profiles seem to vary slightly according to the mucosal anatomical site. For example, *NRAS* mutations are mostly seen in sinonasal forms [30,84], whereas mutations in *KIT*, *NF1,* and *SFB31* are recurrent in anogenital forms [30,85]. Further studies in canine oral melanomas should shed some light on these aspects, revealing recurrent mutations with strong effect in specific locations and specific breeds, reflecting different genetic backgrounds that are easiest to investigate in dog breeds. These data were very recently enriched by comparative expression profiling of 28 cases of oral malignant melanoma, comparing metastatic/primary situations [86].

In both species, the MAPK and PI3K pathways are involved in mucosal melanoma formation and progression [29,30,31,82,85,87]. All of these data suggest that canine and human mucosal melanomas share similarities and that dog could be a good preclinical model for this rare human cancer. 

The search for predisposing mutations has not been extensively investigated in canine melanomas, however, recent analyses of exome sequence data have been performed in human, dog, and horse mucosal melanomas [30]. The authors looked for pathogenic germline variants already established in human melanoma susceptibility genes [88], including *CDKN2A, BAP1, POT1,* and *TP53*, based on their orthologs in the canine genome, but to date none have been identified in canine oral melanoma. 

## 5. Canine Cutaneous Melanoma

Malignant melanoma is a relatively common cancer in dogs and accounts for 9–20% of skin tumors in that species [89]. Interestingly, tumors in cutaneous locations are often benign; 60% of them are benign melanocytomas [34,90], according to pathological criteria, such as nuclear atypias and mitotic index <3 mitoses over ten fields [90,91]. 

In contrast to human cutaneous melanoma for which UV-light exposure is a major etiologic factor, dogs are naturally protected against sun exposure by their hair. Furthermore, black-coated breeds with pigmented skin are at increased risk, as is the case with schnauzers, Scottish terriers, poodles, golden retrievers, dachshunds, cocker spaniels and chow-chows [18,34,42,43,44].

Melanocytic neoplasms in dogs are usually dermal, arising on hair-bearing skin of the head, ears, neck, trunk, and limbs [32,33,34]. Whereas melanocytomas are usually solitary, small, pigmented, firm, and freely moveable over deeper structures, malignant melanomas tend to be fast-growing tumors, often ulcerated and pigmented (Figure 5) [92].

Usually, surgical resection is curative [32,33,34,87,89], although some pathological characteristics, such as a mitotic index ≥3 over ten fields (otf) the presence of ≥20 nuclear atypia, a Ki67 index >15%, ulceration, lymphatic invasion, and tumors extending beyond the dermis, have a negative influence on prognosis [33,35,37,90,91]. More recently, Silvestri et al. showed that Breslow thickness is significantly associated with histological malignancy, clinical outcome, and presence of recurrence/metastasis [93]. 

Cutaneous melanoma has a relatively benign behavior in dogs. Recent studies, evaluating outcomes after surgical excision, showed that the MST was 1363 days and not reached with 77%–88% of dogs alive at 1 year and 54% alive at 2 years [36,39,40,93,94] (Figure 2). In the series of 87 dogs (Laver et al., 2018), 12% of dogs had lymph node metastases and 3.7% had pulmonary metastases at diagnosis. Interestingly, the authors proposed that a mitotic index threshold of 20 mitosis figures over ten fields accurately predicts the outcome. It appears that local resection alone is sufficient for most localized canine cutaneous melanomas, and the clinical benefit of adjuvant therapies like chemotherapy or DNA vaccines has not been shown [40,94]. 

### Genetic Features and Comparative Aspects

The data available on somatic alterations are based on few cases of canine cutaneous melanoma. First, aCGH experiments on five tumors identified copy number alterations such as gains on the CFA 20 and losses on the CFA 6 and 18 [78]. A candidate gene approach with 20 cutaneous melanomas found no variant in *BRAF*, *NRAS*, *PTEN*, *KIT*, *GNAQ,* and *CDK4* [34]. Hendricks et al., (2018) provided genetic results of two cutaneous melanoma cases: an *NRAS* mutation and a translocation in chromosome 10 (region 10–12 Mbases) for one case (Whole Genome Sequencing) and *KRAS*, *TP53,* and *KIT* mutations with amplification of chromosome 30 (region 16,164778 -16,525074 Mbases) for the other case (targeted sequencing) [29]. These preliminary data, together with the fact that canine cutaneous melanocytic tumors are most often benign and not linked to UV-light exposure, confirm that these canine cutaneous forms arise from a distinct mechanism, different from that of most human cutaneous melanomas. Indeed, human UV-induced cutaneous melanomas carry a clear UV-mutational signature and an important mutation burden with recurrent mutated genes such as *BRAF*, *NRAS*, *HRAS*, *NF1*, *CDKN2A*, *TP53,* and *PTEN* and the recently identified genes, such as *PPP6C* and *RAC1* [80,82,95]. However, the identification of genetic alterations in canine cutaneous tumors should provide interesting results and new data for the less investigated non-UV-induced subtypes of melanomas in humans, such as acral melanoma, or rare categories of dermal melanomas [1]. 

In terms of research on human cutaneous melanoma predisposition, numerous GWAS (Genome Wide Association Study) have been performed on large worldwide cohorts of patients and controls over the last 10 years, which has led to the identification of over 20 genome-wide significant melanoma risk regions [30,95,96,97]. While several dog breeds are predisposed to cutaneous melanoma, germline variants are worth exploring to inform the genetics of the rare non-UV-induced subtypes of human melanomas.

## 6. Canine Digital Melanoma

Melanoma is the second most common canine digital tumor after squamous cell carcinoma and accounts for 15%–17% of neoplasias of the digits [98,99]. It arises on the skin digits, footpad, or nail bed and mimics human acral melanoma (plantar surface of the foot, palms of hands, and fingers) (Figure 6). Previous studies showed that between 49–86% of canine melanocytic tumors of the digits were malignant and that all nail bed melanocytic tumors were malignant [34,90,91]. It is a tumor of old dogs, occurring around 10 years of age [98]; some breeds are over-represented like Scottish terrier [98], schnauzer, Beauce shepherds, and Rottweilers [34,99]. 

These melanomas have a biological behaviour intermediate between oral and cutaneous melanomas (Figure 3). They are locally aggressive, with bone lysis occurring in 40–58% of cases [99,100] and have a high metastatic propensity, with regional or distant pulmonary metastasis in 30% to 40% of cases at presentation [99,101]. Without regional or distant metastases, the first-line treatment is amputation of the digit; then, the MST is 12 months, with 42–57% of dogs alive at 1 year and 11–13% alive at 2 years [38,98,99,101]. These data confirm the aggressiveness of digital melanomas, their poor prognosis, and the need for adjuvant therapy. A few studies have reported that chemotherapy does not lead to a survival benefit [40]; xenogeneic DNA vaccine is safe and resulted in an MST of 476 days, but still 50% of dogs treated developed metastases during the course of treatment [100]. 

### Genetic Features and Comparative Aspects

To date, only 3 canine acral tumors have been studied and all harboured *RAS* mutations (2 *KRAS* and 1 *NRAS*), and CNA involving CFA30 was found in one case [29]. In humans, a series of 35 samples of acral melanomas [82] have been shown to present a complex copy number alteration profile more similar to mucosal than cutaneous melanomas [9,82,102]. Significantly mutated genes such as *BRAF*, *NRAS* and *NF1* have been identified in half of the cases [82]. 

Concerning germline variants, there are no data available in dogs or humans, whereas predispositions are known in black-coated dogs from several terrier breeds, and their genetic analyses would greatly inform the rare and severe corresponding human subtypes. 

## 7. Canine Uveal Melanoma 

Melanocytic tumors are the most common primary ocular neoplasms in dogs. They can be limbal or epibulbar, but the most common site is the anterior uvea (iris and ciliary body) [103,104,105]. The biological behavior of these ocular melanomas depends on their location, with uveal melanoma being the most frequent and aggressive [31]. In contrast to humans, primary choroidal melanoma is rare in dogs [32,33]. Between 15 and 29% of canine intraocular melanocytic tumors are histologically malignant [34,37,105], and they are defined by their cytologic abnormalities and a mitotic index superior to four mitosis figures over ten fields [103,106]. More recently, a study tended to discriminate between benign (“melanocytoma-like melanoma”) and malignant uveal melanocytic cases based on morphological criteria such as smaller tumor size, high degree of pigmentation, and low mitotic activity [106]. At examination, the first abnormality commonly observed is a dark-pigmented mass in the anterior segment of the eye [101], but melanoma can be amelanotic in rare cases (Figure 7). Associated clinical signs include iris thickening, irregular pupil with blindness and ocular pain, or secondary symptoms such as keratitis, uveitis, or glaucoma [37,103,104]. 

Only 4% to 8% of malignant uveal melanomas metastasize to lungs and liver, generally within 3 months following enucleation, consistent with what it is seen in localized human iris melanomas [103,107,108]. A few cases with late metastases have been reported in dogs, occurring more than 1.5 years after initial diagnosis [109,110]. Canine intraocular melanomas are treated with enucleation if there is concern about malignancy or complications. In order to preserve the eye and vision, local resection may be performed on isolated primary masses involving only the iris or a portion of the ciliary body, but long-term results are often unsatisfactory. Other available treatments for uveal melanoma include laser therapy, which holds promise for the palliation or potential cure while preserving vision [38].

The prognosis for dogs with histologically benign uveal melanoma (“melanocytoma-like melanoma”) is excellent. However, dogs with malignant uveal melanoma have significantly shorter lifespans [105,106]. In human uveal melanoma, despite successful local treatment, up to 25% and 34% of patients still develop metastases within 5 and 10 years [111]. This situation highlights the importance of genetically discriminating benign and malignant forms. Recently, gene expression profiles (class 1 and 2), Ki67 positivity, and loss of chromosome 3 allowed metastasising from non-metastasising human uveal melanomas to be distinguished [112]. Based on those findings, Malho et al. found that four of these genes (*HTR2B, FXR1, LTA4H,* and *CDH1*) were overexpressed in metastasising canine uveal melanoma [107]. For one of those, *FXR1,* immunohistochemical expression was shown to be different between 10 oral and 10 uveal canine melanoma cases, but the link with the biological behaviour has still to be determined [113]. 

### Genetic Features and Comparative Aspects

In dogs, there are no genetic somatic or germline alterations identified to date. However, in humans, frequent somatic activating mutations in *GNAQ* or *GNA11* have been found, leading to the stimulation of both MAPK and PIK3/AKT pathways [114,115,116]. Another major gene, *BAP1*, has been involved in uveal melanomas, both through somatic alterations and germline variants [117,118]. Moreover, additional risk polymorphisms have recently been identified in human uveal melanomas [119]. Most recently, Robertson et al. (2017) performed an integrated study of 80 uveal melanoma patients, and genetic alterations, such as chromosome 3 monosomy/disomy, alterations in *EIF1AX, SF3B1,* and *BAP-1*, together with CNA and the global DNA methylation status, have been demonstrated to carry a prognostic value [17]. It would be relevant to determine if and how these alterations can discriminate canine uveal “melanocytoma-like melanoma” from malignant melanoma cases.

## 8. Innovative Therapies in Human and Canine Melanomas

In human and dogs, complete surgical excision is the first-line treatment of localized melanomas [38,120,121]. Radiation therapy can improve local control and decrease tumor recurrence after surgery for mucosal melanoma and is effective to treat some types of human cutaneous melanomas, like lentigo maligna melanomas [38,120,121]. However, for advanced stage and metastatic diseases, systemic therapies are essential.

Targeted therapies and immunotherapy have been developed in the last decade and strongly improved some cancer’s management. 

Immunotherapy is a novel treatment modality which consists of treating cancers using and stimulating the patient’s immune system. It has revolutionized some advanced cancer’s treatment and outcome, particularly with the development of checkpoint inhibitors targeting cytotoxic T lymphocytes antigen 4 (CTLA-4) and programmed cell death 1 (PD-1) [122,123]. Those molecules are mainly expressed on activated T lymphocytes, and the binding to their ligands (several clusters of differentiation for CTLA-4 and programmed cell death ligands 1 and 2 for PD-1) suppresses T-cell activity. Upregulation of these inhibitory pathways is observed in many human cancers, particularly melanoma, and is responsible for the suppression of antitumor immune response, which promotes cancer immune evasion. Recently, phase III trials showed that anti PD-1 (pembrolizumab, nivolumab) and anti CTLA-4 (ipilimumab) provided high response rate and durable long-term survival in responding patients, and they are now part of the first-line treatment in advanced stage and metastatic melanomas [122,124]. Checkpoint inhibitors, and particularly the combination of nivolumab and ipilimumab also showed promising results in human mucosal melanomas and should be considered as a good treatment option for advanced stage and metastatic diseases [121,125]. In canine oncology, the tumor microenvironment and immune system are less investigated. However, it has been recently shown that PD-1 and CTLA-4 are significantly overexpressed by peripheral blood T lymphocytes in cancer-bearing dogs compared with in healthy controls [126,127] and that PD-1 blockade enhances T-cell activation [127]. Alongside that, PD-L1 was shown to be expressed in many canine cancers, including oral melanoma [35,128,129]. Those data led to a pilot clinical study in dogs with oral melanoma, using a canine chimeric monoclonal antibody targeting PD-L1. This drug was safe and well tolerated in dogs, and 1/7 dogs showed an objective response [130]. These recent data make the dog an interesting model for translational immunotherapeutic studies that could predict novel drugs efficacy. 

Targeted therapies have also improved outcomes of human metastatic cutaneous melanomas. Those therapies are used according to the molecular profile of the tumors, and cutaneous melanomas with *BRAF* V600 mutations are likely to benefit from the combination of BRAF (vemurafenib, dabrafenib) and MEK inhibitors (trametinib) [124]. Although typically of short duration, antitumor activity with KIT inhibitors like imatinib has been observed in mucosal melanoma harboring *KIT* mutations [121,131]. To date, there are no available targeted therapies against melanoma in dogs. The screening of 21 drugs on nine canine melanomas cell lines showed that they were not sensible to BRAF and KIT inhibitors (vemurafenib, masitinib, imatinib), but to bortezomib, a proteasome inhibitor involved in the NF-κB pathway, that could be a promising targeted therapy [132]. 

## 9. Discussion 

In this review, we attempt to summarize the present knowledge about the different canine melanomas in the context of comparative oncology. 

The dog appears to be the best-known spontaneous model for human oncology, but there are other interesting species like horse, cat, or pig, which can develop melanomas, and genetic studies have been performed in some of these. Advances in our understanding of melanomas have been facilitated by a range of animal models, which contributed to basic biology, characteristics and roles of driver genes, and efficiency of targeted therapies. In horses, melanocytic tumors represent up to 19% of cutaneous tumors and often occur in grey horses [133]. As in dogs, equine melanomas are not thought to be associated with UV-light exposure. Predisposing alterations have been found in grey horses, such as a 4.6kb duplication in *STX17* [134,135]. In cats, melanocytic neoplasms are a rare entity, the most frequent site being the eye; uveal melanoma biologic behaviour is quite similar to human uveal melanoma, because tumors metastasize to internal organs in up to 60% of the cases [38]. However, common mutations present in human uveal melanoma were not found in the feline counterpart [136]. Recently, melanomas of the oral cavity and the skin have also been described, as a retrospective study, in 30 cats [137]. Lastly, the MeLiM pig model is a spontaneous model of cutaneous melanoma, with tumors appearing a few weeks after birth, followed by regression [138]. MeLiM pig genetic data recently led to the identification of new risk loci [139], but no somatic variants have been identified to date. 

At present, there is an opportunity to use these spontaneous models to define new genes and loci that influence melanoma risk, to define recurrent molecular alterations that could be drivers, and to identify therapeutic targets in order to use these data to guide the development of new therapeutic options and achieve better prognosis in humans. 

The great advantage with dogs is the existence of anatomic locations corresponding to those of human melanomas. Canine and human melanomas share many features, such as clinical presentation, pathology, and molecular mechanisms. Particularly, canine oral and digital melanomas display an aggressive behaviour that mimics human mucosal and acral melanomas. However, canine cutaneous melanomas arising on the hair-bearing skin are not linked to UV exposure, contrary to the majority of human cutaneous melanoma, and thus might represent relevant models for the rare non-UV-induced cutaneous melanomas, but not for the classical SSM forms of human cutaneous melanomas. 

To date, genetic studies of canine melanomas, mainly based on oral subtypes, showed that their genomic landscapes are similar to those reported in human non-UV-induced melanomas, with a low point mutation frequency but a high structural variation burden. Those features, as well as the absence of UV signature make the dog a good opportunity to study any form of non-UV-induced human melanomas (cutaneous and noncutaneous).

Another advantage of the dog is the existence of breeds that constitute genetic isolates. Some breeds are at higher risk to develop melanomas, such as breeds with pigmented skin or mucosa, and are ideal spontaneous models to screen for pathogenic germline alterations. Moreover, as there is a likely association between the specific genetic background of breeds and the landscape of somatic mutations, dog can help to identify relevant alterations in human melanoma subtypes. 

Genetic analyses in dogs such as whole genome sequencing, exome, RNA sequencing, or a CGH are now routinely performed. Canine sample collection is facilitated by the prevalence of melanoma in predisposed breeds, and thanks to the efforts of the community. Biological Resource Centres have been set up, like the French Cani-DNA BRC (http://dog-genetics.genouest.org) or the American Canine Comparative Oncology Genomics Consortium (CCOGC) (https://ccogc.net/), which have significant collections of canine melanoma samples. 

Those resources and methods can be used to identify therapeutic targets and enhance clinical trials in pet dogs [35]. This concept of “first-in dogs trials” has been launched by the National Cancer Institute’s (NCI’s) Comparative Oncology Trials Consortium (COTC; http://ccr.cancer.gov/resources/cop/COTC.asp), with the aim of answering biological questions that can inform the development of novel agents for future use in human cancer patients [140,141]. Clinical trials in pet dogs offer several advantages. First, they are not constrained by traditional Phase I, Phase II, and Phase III trial designs, and make it possible to address many questions in one study (toxicity, pharmacokinetics, pharmacodynamics, response etc.) to inform future human clinical studies. Moreover, dogs present a shorter lifespan and a relatively rapid disease course for most cancers, thus providing the cancer natural history with earlier outcome measures compared with those from human patients. Since many canine cancers do not have well-established standards of care, novel agents can be offered to pet dogs alternatively to conventional therapies, leading to trials benefiting the health of both dogs and humans.

## Figures and Tables

**Figure 1 genes-10-00501-f001:**
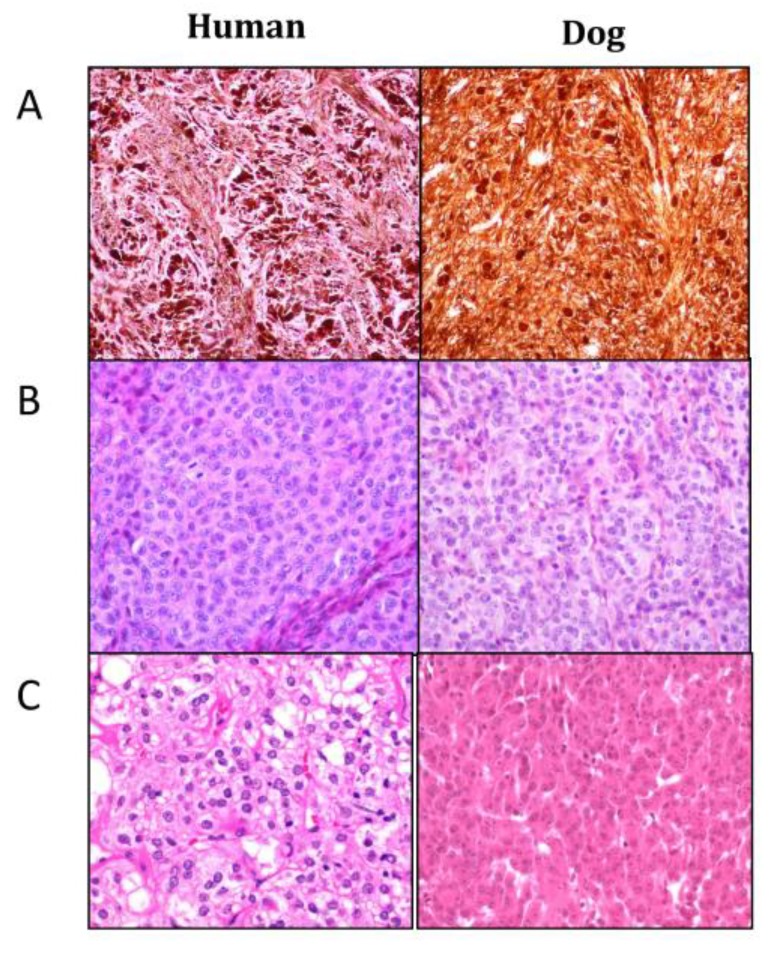
Histopathological aspects of human (left) and dog (right) melanoma morphological subtypes. Hematoxylin–Eosin–Safranin (HES) staining of formalin-fixed and paraffin-embedded biopsies. Photo credit: B. Vergier and J. Abadie. (**A**) “Animal type”: sheets or bundles of highly pigmented large epithelioid to spindle-shaped neoplasic melanocytes; (**B**) “Simulating (congenital or cellular blue) naevi type”: sheets with a high cellular density of small oval to spindle cells with a small central nucleus containing one to several small nucleoli; (**C**) “Epithelioid type”: large round to polygonal epithelioid cells with a high amount of variably pigmented cytoplasm and a large nucleus.

**Figure 2 genes-10-00501-f002:**
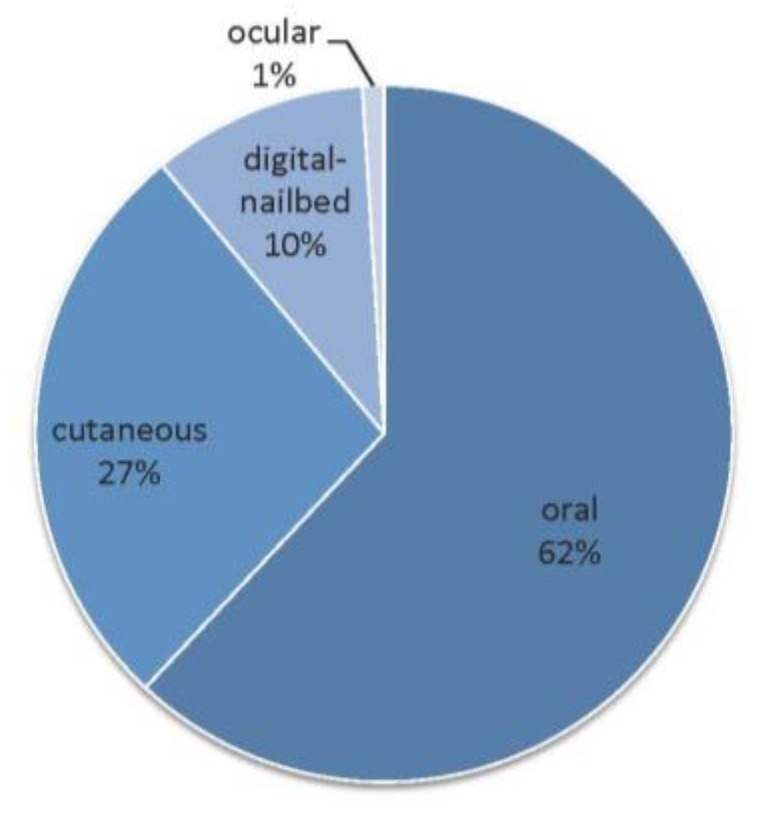
Distribution of canine malignant melanomas (n=1652) depending on the anatomical site [34].

**Figure 3 genes-10-00501-f003:**
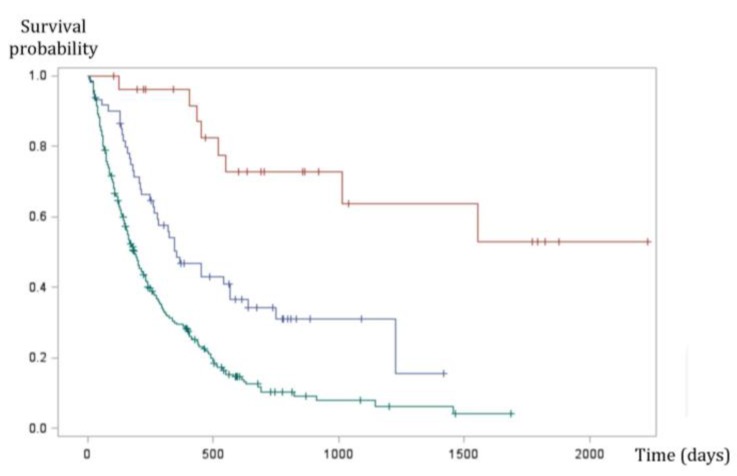
Kaplan Meier survival curve of 335 dogs with malignant melanoma. Oral (green line), digital (blue line), and cutaneous (red line) melanomas have significantly different prognosis (*p* < 0.0001, log-rank test). Oral melanomas have the worse prognosis with a MST (median survival time) of six months, whereas survival times were highest in cutaneous forms (MST not reached). Digital forms seem to have intermediate prognosis with an MST of 11.8 months (Prouteau et al., in preparation [39]).

**Figure 4 genes-10-00501-f004:**
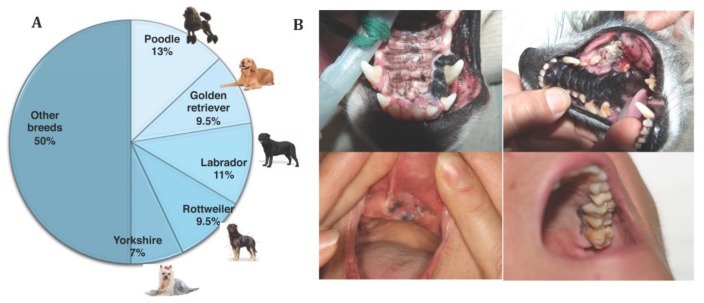
(**A**) Breed distribution in a study of 247 canine oral melanomas. Five breeds represent 50% of cases, illustrating the breed predisposition (Prouteau et al., in preparation [39]). (**B**) Pictures of canine oral melanoma involving the gingiva; that can be pigmented (on the top left) or amelanotic (on the top right). Pictures of human mucosal melanoma (pigmented tumors) arising on the gingiva of the oral cavity. Photo credit: P. De Fornel, A. Dupuy and T. Jouary.

**Figure 5 genes-10-00501-f005:**
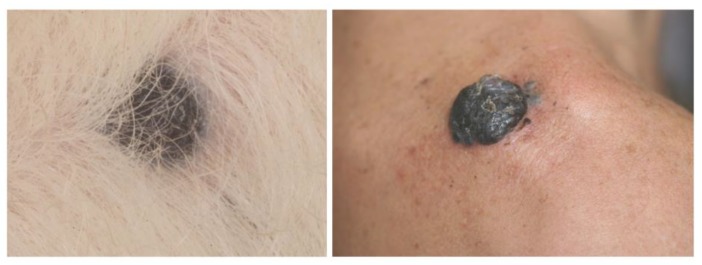
Canine cutaneous melanoma (left) and human nodular melanoma (right). Photo credit: P. Durieux and T. Jouary.

**Figure 6 genes-10-00501-f006:**
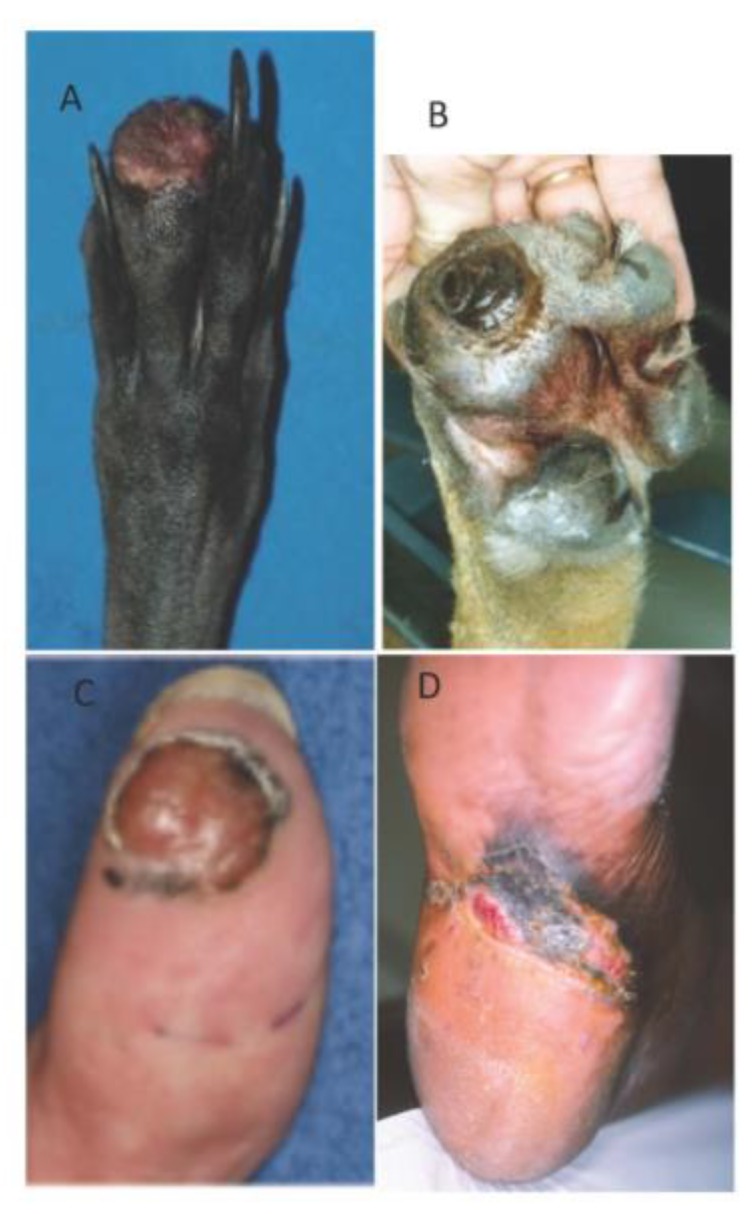
Canine digital melanomas mimic human acral melanomas. Canine digital melanomas can occur on the digit (**A**) or the footpad (**B**) and are often ulcerated. Human acral melanomas typically occur on the palmar/plantar surface of the hand/feet (**C**,**D**). Photo credit: A. Muller (**A**), F.A. Fogel (**B**), A. Dupuy (**C**), and E. Maubec (**D**).

**Figure 7 genes-10-00501-f007:**
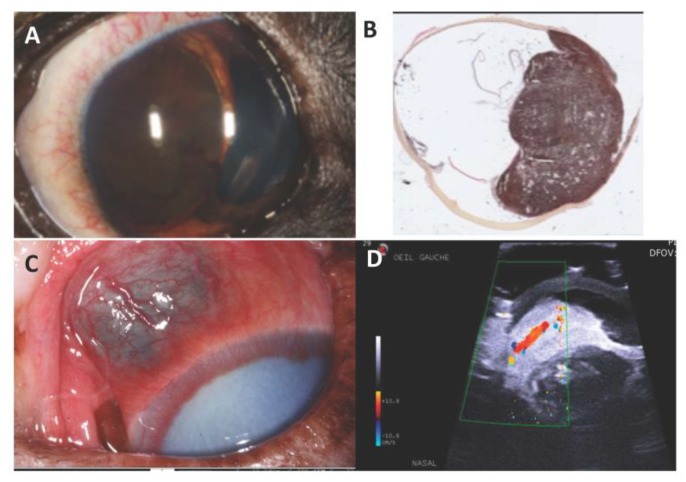
Canine uveal melanoma: (**A**) dog eye with an iridal melanoma; (**B**) Histological section of a canine uveal melanoma; (**C**) Melanoma of the ciliary body in a dog. Localized dilation of the episcleral vessels indicating, as in human, a melanoma of the ciliary bodies; (**D**) Ultrasonography of an affected eye. We notice the high surface echogenicity with rapid decay of this echogenicity Photo credit: P. Durieux.

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
