# Peer review of "Canine Melanomas as Models for Human Melanomas: Clinical, Histological, and Genetic Comparison"

_genes, 2019, doi:10.3390/genes10070501_

Round 1
Reviewer 1 Report
The authors wrote a review about dog melanoma as a model for comparative oncology. They describe different forms of melanoma, and clinical, pathological and genomic issues related to each subtype of this cancer.
This review is very detailed and contains much information about canine melanomas and clinical and pathological aspects. However, a few concerns need to be addressed in order to improve the manuscript.
Broad comments
- In general, only a few words through the manuscript are dedicated to the therapies used in the clinics, and to a comparison between therapies used in veterinary medicine and for humans. For example, it would be interesting to read a few words about checkpoint inhibitors and their targeting in human, as well as what has been found in dogs concerning those molecules. Are there any other targeted therapies used in dogs?
Also, the resistance mechanisms frequently encountered in human melanoma are not mentioned. For bad prognosis canine melanomas, resistance should be an issue to investigate/to discuss.
- More details on the predisposition genetics and data on etiology should be found in the introductory part. As an example for introduction and also for discussion, the authors wrote several times that melanoma occur preferentially on black or dark coated animals, which is opposite to humans, where people with fair skin are at higher risk and with MC1R RHC variants increasing this risk.
- Line 100: In addition to genetic differences between mouse and human, one should add biological differences in melanocytes from both species. Also, there is no clear discussion about the dermal position of melanocytes in dogs, whereas melanocytes are located on the basal layer of the epidermis in human. Cells are therefore in different microenvironments, which is determining for tumor initiation and progression. It is also crucial for local immune response. These issues should be discussed.
- Line 207: is human oral melanoma also radiosensitive, contrary to cutaneous melanoma? This should be discussed
- Line 233: Again those criteria should be discussed in the light of human pathology.
Specific comments
- Lines 51-52: please indicate the prognosis associated with different subtypes
- Line 71: “different from cutaneous subtypes and a much worse prognosis”. Later in the paragraph, a five year survival rate between 25 and 33% is mentioned for mucosal melanoma. So what is the equivalent measure for cutaneous melanoma and is it really better?
- Lines 186-190: this paragraph should be moved to a more general paragraph describing canine melanomas (starting line 117 for example)
- Line 304:“Local resection is sufficient”. Do you mean it is sufficient when the tumor is till localized and not yet metastasized?
- Line 401: the link between the two studies should be clarified.
Author Response
Prouteau and André: point by point responses to the reviewers
As mentioned by both reviewers, the English language needed minor spell check, what we did through out the paper.
The addition and edits required from the reviewers are indicated in red in the manuscript.
1) In general, only a few words through the manuscript are dedicated to the therapies used in the clinics, and to a comparison between therapies used in veterinary medicine and for humans. For example, it would be interesting to read a few words about checkpoint inhibitors and their targeting in human, as well as what has been found in dogs concerning those molecules. Are there any other targeted therapies used in dogs? Also, the resistance mechanisms frequently encountered in human melanoma are not mentioned. For bad prognosis canine melanomas, resistance should be an issue to investigate/to discuss.
We added a paragraph in the conclusion, entitled « Innovative therapies in human and canine melanomas » to discuss immunotherapy and targeted therapies in humans and dogs, lane 437-476.
2) More details on the predisposition genetics and data on etiology should be found in the introductory part. As an example for introduction and also for discussion, the authors wrote several times that melanoma occur preferentially on black or dark coated animals, which is opposite to humans, where people with fair skin are at higher risk and with MC1R RHC variants increasing this risk
Sure, more explanations have been included in the introduction section lanes 132 to 144.
3) Line 100: In addition to genetic differences between mouse and human, one should add biological differences in melanocytes from both species. Also, there is no clear discussion about the dermal position of melanocytes in dogs, whereas melanocytes are located on the basal layer of the epidermis in human. Cells are therefore in different microenvironments, which is determining for tumor initiation and progression. It is also crucial for local immune response. These issues should be discussed.
Yes, the location of melanocytes and more importantly the location of the melanoma is an important issue. In fact, the non-UV characteristics of canine melanomas (both oral/mucosal and cutaneous) led to the fact that the tumorigenesis mechanism is not the transformation of epidermal melanocytes into a tumour, due to UV damages (direct exposition), but to other genetic and/or environment and microenvironment features. In fact, melanocytes and melanocyte precursors, both in dog and in human skins are located at both places : in the basal layer of the epidermis and in the dermis. In dogs, of course, the presence of melanocytes in the numerous hair follicles of the dermis is to be taken into consideration but is not fully explored so far.
We didn’t add this notion in the chapter lane 100, because this chapter present the broad similarities between dog and human cancers in general; but we added few sentences about the melanoma/melanocytes location and the UV-induced / non-UV-induced status of human and dog melanomas in the Canine melanoma chapter, lane 132-139.
4) Line 207: is human oral melanoma also radiosensitive, contrary to cutaneous melanoma? This should be discussed.
Yes it has been showed that radiationtherapy improve local control and decreases recurrence rate in mucosal melanoma but no proof that it improves overall survival. Radiation therapy in human is mentionned at the begining of the paragraph « Innovative therapies in human and canine melanomas » lane 439-442
5) Line 233: Again those criteria should be discussed in the light of human pathology.
Yes, this has been done lane 250-253.
Specific comments
6) Lines 51-52: please indicate the prognosis associated with different subtypes
We aded it in the « cutaneous melanoma paragraph » in the introduction, lanes 63-68. « Acral subtypes carry the worse prognosis among cutaneous melanomas, with a 5-year survival rate around 80% (Criscito et al 2017). The impact of other subtypes on prognosis is not known (Greenwald 2012). »
7) Line 71: “different from cutaneous subtypes and a much worse prognosis”. Later in the paragraph, a five year survival rate between 25 and 33% is mentioned for mucosal melanoma. So what is the equivalent measure for cutaneous melanoma and is it really better? :
For cutaneous melanomas of all stages, the 5 year survival rate is around 90%, so it is really better than mucosal melanoma; we added it in the text, in the introduction, cutaneous melanoma part, lane 65-68.
8) Lines 186-190: this paragraph should be moved to a more general paragraph describing canine melanomas (starting line 117 for example) :
We moved it and also the associated Figure to the chapter « Canine melanoma » in the introduction, lane 145. Consequently, the numbering of the figures has been changed.
9) Line 304:“Local resection is sufficient”. Do you mean it is sufficient when the tumor is till localized and not yet metastasized?
Yes, we modified the text lane 322: « It appears that local resection alone is sufficient for most localized canine cutaneous melanomas, and the clinical benefit of adjuvant therapies like chemotherapy or DNA vaccines has not been shown ».
10) Line 401: the link between the two studies should be clarified:
We changed the sentences to clarify line 415-421

Reviewer 2 Report
The manuscript entitled "Canine melanomas as models for human melanomas: clinical, histological and genetic comparison” presents an exhaustive review of the comparative value of dog melanoma for our understanding of the human disease. The article is excellently written and adequately presents relevant information in a well-organized way.
Perhaps, the authors can consider to insert some more citations of recent reviews for the single subtypes of melanoma that can guide the non-expert reader to more in depth information.
Author Response
Prouteau and André: point by point responses to the reviewers
As mentioned by both reviewers, the English language needed minor spell check, what we did through out the paper.
The addition and edits required from the reviewers are indicated in red in the manuscript.
1) Perhaps, the authors can consider to insert some more citations of recent reviews for the single subtypes of melanoma that can guide the non-expert reader to more in depth information
We added a reference (Greenwald et al 2012) (ref 5) about SSM and nodular melanoma, in the introduction, lane 53.

Round 2
Reviewer 1 Report
The authors properly answered the comments raised.